# High-Resolution Lacustrine Records of the Late Holocene Hydroclimate of the Sikhote-Alin Mountains, Russian Far East

**DOI:** 10.3390/biology12070913

**Published:** 2023-06-26

**Authors:** Nadezhda Razjigaeva, Larisa Ganzey, Tatiana Grebennikova, Tatiana Kopoteva, Mikhail Klimin, Khikmatulla Arslanov, Marina Lyashchevskaya, Alexander Panichev, Sergey Lupakov

**Affiliations:** 1Pacific Geographical Institute, Far Eastern Branch of the Russian Academy of Sciences, Radio St., 7, 690041 Vladivostok, Russia; lganzey@tigdvo.ru (L.G.); tagr@tigdvo.ru (T.G.); lyshevskay@mail.ru (M.L.); sikhote@mail.ru (A.P.); rbir@mail.ru (S.L.); 2Institute of Water and Ecological Problems, Far Eastern Branch of the Russian Academy of Sciences, Dikopoltsev St., 56, 680000 Khabarovsk, Russia; kopoteva@ivep.as.khb.ru (T.K.); m_klimin@bk.ru (M.K.); 3Institute of Earth Sciences, St. Petersburg State University, Universitetskaya Nab., 7/9, 199034 St. Petersburg, Russia; arslanovkh@mail.ru

**Keywords:** mountain lake/mire complexes, diatoms, botanical composition, inundation and dry periods, monsoon, cyclogenesis

## Abstract

**Simple Summary:**

We studied the sediments of landslide-formed small mountain lakes (elevation 565 and 750 m a.s.l.) on the slopes of an ancient volcano. The lakes are located in the territory of the Sikhote-Alin Biosphere Reserve, with unique ecosystems that combine warm- and cold-climate vegetation and where rare animals live. These lakes are natural archives, allowing the reconstruction of the natural environment of this unique area with a temporal resolution of up to 30 years. Nizhnee Lake, which developed over a period of 2640 years, is the most sensitive to climate change. Ten stages of the lake’s evolution, with periods of watering and swamping, were identified and controlled by precipitation. Our study demonstrates how the communities of swamp plants and diatom microflora responded to temperature fluctuations associated with solar activity and regional manifestations of global climatic events. The regional drivers were the intensity of summer and winter monsoons, and the change in the positions of the atmosphere action and cyclonic activity centers. The natural environment was especially unstable during the Little Ice Age (14th–19th centuries), which was wet with short-term dry episodes.

**Abstract:**

There is little information about moisture changes in different altitudinal belts in mountainous regions of the southern Russian Far East. We present ecological and taxonomic compositions of the diatom flora and identify the botanical composition of peat in small mountain lake/mire complexes located in the Central Sikhote-Alin Mountains, within large landslides on the paleovolcanic slopes. Frequent changes in diatom assemblages and peat-forming plants indicate unstable hydroclimatic conditions with varying degrees of wet and dry conditions up to the overgrowth of the lakes. Frequent change in sphagnum mosses with different trophic preferences was identified. The chronology is based on 11 radiocarbon dates. Accumulation rates reached 1.7–1.9 mm/year, and the temporal resolution for the reconstructions was up to 30–40 yr. The tendencies of lake evolution depended on different scale hydroclimatic changes over the last 4400 yr. The most detailed data for the last 2600 yr were obtained from the Nizhnee Lake sequence, which is more sensitive to climatic changes. The main reason for the change in the hydrological regime of the lakes was variations in precipitation during short-term climatic changes. The sediment record of moisture fluctuations is relatively well correlated with regional patterns reflecting summer monsoon intensity and cyclogenesis activity.

## 1. Introduction

The climatic instability of recent decades makes paleoclimate research more relevant. Paleorecords make it possible to assess the scale of climate change and the response of biotic components to warming and cooling in the past. One of the East Asian climatic features is uneven moisture supply throughout the year, controlled by the intensity of winter and summer monsoons [1,2,3,4]. The specificity of atmospheric circulation here is determined by the interaction of air masses in the ocean–atmosphere–continent system. The climate is greatly influenced by cyclonic activity resulting from the interlatitudinal exchange of air masses. The climatic situation controls the seasonal irregularity of moisture, which is associated with the passage of extreme floods and long droughts. The duration of periods with different humidity varied significantly during the Holocene. Winter monsoon variability features exerted significant impacts on Eurasian and Western Pacific climatic patterns during this epoch [1]. The dynamics of the summer monsoon during the Holocene are well-studied for northeast China, as well as the Korean Peninsula and the Japanese Islands [2,3,4,5,6,7,8]. For these regions, there are chronicles informing on the hydroclimatic changes during the Common Era [5,9,10].

In the Southern Russian Far East, regular instrumental meteorological observations began in the second half of the 20th century; therefore, on a scale of several centuries–millennia, climatic trends and the intensity of cyclogenesis can only be estimated on the basis of paleoenvironmental records. There is little information on moisture level variations at different altitudinal belts and the response of biota to the changes. Such studies are especially important for mountain areas with unique ecosystems that combine heat-adapted and cold-adapted plants, including endemics. The strong microclimatic variability in the mountains contributed to refugia preservation, which played an important role in changing areas and altitudinal belt boundaries during climate changes [11].

Reconstructions performed for different regions of the Southern Far East showed that the development of landscapes during the Holocene was determined mainly by hydroclimatic changes associated with the intensity of the East Asian monsoon and the activity of cyclogenesis [12,13,14]. Lacustrine sediments are the most informative for studying high-resolution records, making it possible to restore the response of the biota to moisture changes. In Primorye, most of the climatic reconstructions were created using coastal lake sediments [15,16,17,18], mountain plateau paleolake sediments of East Manchurian Mountains [14] and the Southern Sikhote-Alin [19,20], and a paleolake in the river valley [21]. These studies highlight the dynamics of vegetation and lake microflora development, which are closely related to changes in hydroclimatic conditions during the late Holocene. For the Central Sikhote-Alin, reconstructions have been created only for the Izyubrinye Solontsi Lake from the group of Solontsovskie (Shanduyskie) lakes [22]. These lakes are located in the Sikhote-Alin Biosphere Reserve, included in the UNESCO World Heritage Site, with the mission to preserve biological and landscape diversity, and to monitor the Central Sikhote-Alin ecosystems. The group of Solontsovskie lakes includes unique water bodies located at different heights above sea level; data on them make it possible to determine the dynamics of biota at different altitudinal zones.

The objectives of our study were (1) to reconstruct the environmental and climatic changes in the Central Sikhote-Alin Mountains using multi-proxy records of the Nizhnee Lake sediments, and (2) to compare our data with local and regional paleoclimatic data in order to provide new information on paleoclimate variability in poorly studied mountainous areas with diverse ecosystems in the Russian Far East.

## 2. Regional Setting

The Solontsovskie lakes are located on the eastern macroslope of the Sikhote-Alin, near the paleovolcano Solontsovsky, which erupted ~61–56 Ma [23]. There are 10 lakes formed within large landslides that block the stream channels. We investigated two of these lakes: Nizhnee and Izyubrinye Solontsi. The newly studied record of the Nizhnee Lake is presented in this manuscript. Paleoclimatic records from Izyubrinye Solontsi Lake sediments have been published elsewhere [22]. The Nizhnee Lake (100 × 50 m) is located at 565 m a.s.l., within the altitudinal zone of Korean pine and Korean-pine-spruce forests (400–700 m), and the Izyubrinye Solontsi Lake (190 × 100 m) is located at 750 m a.s.l, within fir–spruce forests (700–1200 m). Distance to the seacoast is 33 km (Figure 1).

The average annual temperature in the mountains at an altitude of 500–600 m is +1.5 °C; the temperature in January is –18.3 °C, and in August, +11.8 °C [24]. The amount of precipitation is 940 mm/year, with the maximum in August–September. The frost-free period averages 120 days. The height of the snow cover reaches 1 m. Data from the nearby Terney station (WMO 31909) were used in this study.

The features of regional moisture circulation determine the hydrological regime of the Solontsovskie lakes [22]. At the beginning of the summer, water often almost disappears due to a high rate of evapotranspiration that leads to filling up by vegetation. During the typhoon season (August–September), the lakes are filled with rainfall, the water level rises promptly, and open water appears. The lakebeds are composed of permeable deposits, and the lake basins are almost completely infilled with peat.

The sod in the marginal part of the Nizhnee Lake basin consists of herb remains of grasses (abundance 65–90%), with a predominance of cottongrass (*Eriophorum scheuchzeri, E. vaginatum*), sedges (*Carex criptocarpa, C. middendorffii*), *Rynchospora alba*, *Schoenoplectus tabernaemontanii*, fern *Dryopteris thelypteris, Iris* sp. Sphagnum mosses (abundance 10–35%), eutrophic *Sphagnum denticulatum, S. subsecundum*, as well as hygrophyte *S. divinum*, typical for oligotrophic–mesotrophic heavily watered bogs, and green moss (≤1%). The community of sphagnum mosses allows us to conclude that the swamp belongs to the mesotrophic type. Larch (*Larix cajanderi*) grows along the edges of the lakes. At present, the Izyubrinye Solontsi Lake turned into a mesotrophic swamp dominated by *S. divinum* and cottongrass Eriophorum polystachion with cranberry *Oxycoccus palustris*, *Chamaedahpne calyculata*. There are also Calamagrostis angustifolia, sedge Carex rhynchophysa, Drocera rotundifolia, Menyanthes trifoliata, and Glyceria spiculosa. In the frame of the swamp grows larch *Larix cajanderi*.

## 3. Materials and Methods

The marginal part of the Nizhnee Lake (45°25′23.56″ N, 136°30′58.54″ E) was drilled using a Russian peat corer. The length of the sampled section was 3.25 m depth. The section consists of dark brown peat, composed of decomposited remains of plant material. We took 65 samples of the peat at 5 cm intervals without gaps and stored them in sealed plastic bags. Paleoclimatic records from Izyubrinye Solontsi Lake sediments have been published [22], and we use these data to compare the evolution of Lake Nizhnee, which gives a more general picture of environmental change in the mountains.

The vegetation comprising the peat was determined according to Kulikova [25]. Charcoal fragments, green algae, cladocera, and testate amoebae were also recorded. Testate amoebae were identified according to Mazei and Tsyganov [26]. The decomposition of plant remains was assessed as the proportion of unidentifiable humic substance [25]. Ash content (nonorganic content, %) was determined after ignition at 550 °C [27]. The ash content of a peat sample is the percentage of dry material that remains as ash after combustion.

Diatom samples were prepared using procedures described by Gleser et al. [28] and Battarbee [29]. The samples were heated to 100 °C with 30% hydrogen peroxide (H_2_O_2_). The diatom samples were mounted on permanent slides; 0.2 mL of the suspension was pipetted onto a clean coverslip, and then the water was evaporated at room temperature. After the coverslips were dry, they were then mounted in El’yashev aniline–formaldehyde resin with a refractive index of 1.67–1.68 [28]. Identification of diatoms was performed with a microscope Axioscop (Carl Zeiss, Oberkochen, Germany) at ×1000 magnification. More than 300 valves were counted on each slide. We used atlases of diatoms for species identification and environmental characteristics of individual species [30,31,32,33,34,35]. The content of diatom valves in 1 g of air-dry sediment was calculated to determine the productivity of diatom flora depending on the ecological state of the lake at different stages of its development. The main taxa were indicative of changes in environments, which are shown in the percentage diagram (abundance ≥ 5%). The diatom record was divided into assemblages based on cluster analysis. Principal Component Analysis (PCA) was used as additional proof of zonation. Cluster analysis and PCA was performed using PAST 3.26 software [36]. To conduct a factor and cluster analysis, we used dominant and subdominant diatoms; rare species occurring ≤5% relative abundance were removed. The taxonomic composition of diatom communities was assessed by the Shannon index (H). Traces of desolution in valves are absent.

Radiocarbon dating of bulk peat samples was carried out (beta counting) at the Institute of Earth Sciences, St. Petersburg State University (St. Petersburg). Radiocarbon dating of peat was carried out by benzene synthesis methods. The peat samples were treated with 2% HCl and NaOH to remove carbonates and foreign humic acids. The samples of well-decomposed peat were dated using hot 2% NaOH applied to the humic acid extract. Calibration of radiocarbon dates was performed using the OxCal 4.4 and “IntCal20” calibration curve [37,38]. The calibrated age was determined by the age-depth model constructed using Bacon 2 [39].

## 4. Results and Interpretation

### 4.1. Ages, Sedimentation Rates, and Resolution

The age-depth model created on the ^14^C-dates shows that peat accumulation rates in the Nizhnee Lake were fairly uniform (Figure 2, Table 1). According to the age-depth model, the base of the borehole is estimated at 2640 years. At the initial stage, the peat accumulation rate was 0.8–1 mm/yr. Rates increased to 1.6–1.7 mm/yr~1290 years ago, decreased to 1.4 mm/yr 980–620 cal yr BP, increased ~620–320 cal yr BP (1.6–1.7 mm/year), and decreased in the final stage (to 1.2–1.4 mm/year). The approximate resolution of the paleoenvironmental reconstructions is 50–60 years, and 30–40 years for the last 1290 years.

The rates of peat accumulation in the Nizhnee Lake were lower than those in the Izyubrinye Solontsi Lake, where the rate was up to 1.9 mm/year [22], but general trends were consistent. A sharp increase in the rate occurred ~1280 years ago, at the end of the Medieval Warm Period, and at the beginning of the Little Ice Age, the rate decreased, then increased ~500–400 cal yr BP, and has been in decline for the last 400 years.

### 4.2. Mire Vegetation Development Based on Botanical Composition of the Peat

Waterlogging began in a depression within the landslide, occupied by a larch forest. Under humid and cool conditions (2640–2590 cal yr BP), grass–green moss peat began to accumulate (Figure 3). Remains of five species of green mosses were found: *Aulacomnium palustre, Helodium blandovii, Mnium punctatum, Meesia trifaria,* and *Drepanocladus* sp. Herbaceous plants are represented by sedges, cottongrass, reeds, and iris. The sphagnum cover included the hygrophytes *Sphagnum jensenii, S. divinum,* and *S. fallax*. A larch forest with birch was developed along the border of the bog up to 2430 cal yr BP, and eutrophic sphagnum grass and woody grass peat were formed (325–305 cm). The presence of a large number of burnt twigs and charcoal pieces indicates frequent fires. The cladocera *Chydorus sphaericus*, a species widely distributed in eutrophic freshwater shallow water bodies, appeared. In 2590–2430 cal yr BP, large-scale fires periodically occurred that caused mire vegetation destruction, as evidenced by the presence of microcharcoals and green moss *Aulacomnium palustre*, an indicator of pyrogenic successions [40].

Lowland grass peat (305–280 cm), mainly cotton grass sedge, with a significant presence of hydrophytes, including lake cane (*Scirpus lacustris*), horsetail *Equisetum fluviatil*, and water lily *Nymphaea alba*, began to accumulate ~2430–2160 cal yr BP. The shrub layer was poorly developed; remains of shrub birch were found. Sphagnum and green mosses practically disappeared (2330–2220 cal yr BP). The degree of decomposition of the material decreased. Peat formed ~2380–2110 cal yr BP has an increased mineral content (up to 43.5%).

The moss cover became more developed ~2160–1290 cal yr BP; the remains of sphagnum mosses reach up to 50%, and green mosses up to 25% (depth 280–200 cm). Among the herbs, cottongrass and sedge prevailed, the remains of *Comarum*, lobelia, and reeds were found, and there were a lot of irises. Cranberries (*Oxycoccus palustris*) appeared in abundance. The degree of decomposition decreased. The presence of cladocera indicates waterlogging.

The greatest flowering of sphagnum mosses (*Sphagnum jensenii*) was ~1290–1260 cal yr BP (depth 200–195 cm). *Baeothryon* sp. appeared among herbs. The outbreak of eutrophic hygrohydrophyte green moss *Limprichtia revolvens* (depth 195–185 cm) indicates a stagnant water regime 1260–1200 cal yr BP.

Herbaceous moss peat (depth 185–155 cm) was formed 1200–1010 cal yr BP. Along with *Sphagnum jensenii,* the role of *S. divinum* has increased since 1100 cal yr BP. There are single charcoals, with the exception of an interval of 165–160 cm (1080–1040 cal yr BP). The number of sphagnum and green mosses decreased ~1010–980 cal yr BP, and herbaceous peat (depth 155–150 cm) with an abundance of cottongrass remains began to form, possibly as a result of frequent fires. Herbaceous sphagnum peat (depth 150–85 cm) of a rather homogeneous composition accumulated ~980–560 cal yr BP. Green mosses are represented by *Limprichtia revolvens* and *Meesia trifaria*. Cranberries grew in abundance, and many larch needles were found. A high abundance of cladocera *Alonopsis elongate* indicates a wet environment.

The role of herbaceous plants increased 560–210 cal yr BP (depth 85–35 cm). Among the sphagnum mosses, the mesotrophic *Sphagnum lindbergii* and the eutrophic hydrohygrophyte *S. subsecundum* appeared. The cladocera practically disappeared ~440 yr BP when conditions became drier. Herbaceous peat formed ~380–350 cal yr BP. The amount and diversity of green mosses (*Limprichtia revolvens, Scorpidium scorpioides,* and *Calliergonella cuspidata*) increased ~320–210 cal yr BP. The sphagnum moss layer became more developed at 210–130 cal yr BP (depth 35–25 cm). There was a cottongrass sedge mire ~130–90 cal yr BP. The role of sphagnum mosses increased in recent decades, but the surface sod is composed mainly of grasses.

In peat formed 1870–500 cal yr BP (255–80 cm), testate amoebae are present in large numbers: hydrophiles *Heleopera petricola* and *Archerella flavum* [41], hygrophilic sphagnophiles *Hyalosphenia papilio*, and occasionally *H. elegans*, which are typical of a mesotrophic environment [42]. The last two species have an optimum level of swamp waters of 20–21 cm and are tolerant to fluctuations of 5–15 cm [43]. In peat younger than 2220 years (from a depth of 285 cm), *Centropyxis aculeata, C. constricta,* and *C. laevigata* are found; these species are typical of wet habitats and tending to mesotrophic environments [41].

### 4.3. Lake Development Based on Diatom Data

A total of 144 species of diatoms were identified, with epiphytes (69) and bottom forms (64) predominating, and 11 of them being planktonic and temporary planktonic. Most of the identified diatoms are cosmopolitan, 11 species are boreal, and 22 are arctoboreal species. A total of 128 species, all oligohalobes, have been identified based on their attitude to mineralization. With reference to mineralization, indifferent species are more prevalent (78), followed by halophobes (39) and halophiles (11). The dominant group is circumneutral species (66) relative to pH; 40 species are acidophiles, and 24 are alkaliphiles. The diatom distribution suggests 10 zones reflecting the lake’s evolution and changes in the environment (Figure 4).

The clusters distinguish the diatom zones from the lower and the upper parts of the section (zones 1–3 and 5–10). However, the clustering of diatom zone 4 was not well pronounced, and we used percentage proportions. The sharp increases in *Aulacoseira laevissima,* planktonic species typical for shallow lakes, allowed us to draw a border at this level. Shannon indexes decrease in the case of strong domination of one diatom species, and variation reflects an unstable regime (zones 6–9).

At the initial stage (2640–2330 cal yr BP, depth 325–295 cm), the lake had a littoral zone with aquatic plants. Planktonic species prevailed (zone 1) (Figure 5). *Aulacoseira subarctica* and *Aulacoseira italica*, typical for mesotrophic eutrophic waters, dominated, and among epiphytes, *Staurosira venter*, typical for oligotrophic, mesotrophic conditions [34], dominated. With respect to salinity, indifferent species predominate; with respect to pH, circumneutral species and alkaliphiles predominate. The concentration of diatom valves varies and reaches 46.9 × 10^6^ valves/g.

A short-term decrease in the lake level occurred during the cooling 2330–2280 cal yr BP (depth 295–290 cm). Diatom zone 2 is characterized by a high content of benthic flora. Dominants are temporarily planktonic cosmopolitan *Tabellaria fenestrata*, the arctoboreal *Fragilariforma nitzschioides*, which prefers oligotrophic dystrophic waters, and the benthic *Hantzschia amphioxys* (13.2%), that can also inhabit soil. The subdominants are the benthic *Pinnularia crucifera*, typical for oligotrophic lakes of the northern regions, and the swamp *Eunotia glacialis.* The diversity of diatom flora increased, along with the content of halophobes (up to 41.3%), acidophiles (up to 36%), and arctoboreal species (up to 15.5%). The lake became closer to oligotrophic, and the temperature conditions, apparently, became colder. The concentration of valves decreased to 2.9 × 10^6^ valves/g. The depression began to overgrow, and in some places, dry areas probably formed.

The subsequent watering of the lake basin (2280–2110 cal yr BP, depth 290–275 cm) led to the predominance of planktonic species (up to 79.6%) (zone 3). *Aulacoseira subarctica* and *A. valida*, common in oligotrophic, mesotrophic water bodies of northern and mountainous regions, dominated. The peak in these species coincides with the rapid increase in diatom concentration, indicating the rise in diatom productivity. *A. subarctica* prefers heavy snowfalls and thick ice, and the development of this species depends on the amount of winter precipitation and the length of the ice cover [44]. In Kamchatka, this species grew when the water temperature was ≤4 °C; it may develop in low light conditions [45]. The subdominant is *Staurosira venter*. The proportions of acidophyles and halophobes decreased. The content of arctoboreal species is ≤8.1%. The concentration of valves is 31–70 million/g. The composition of diatoms indicates the existence of an oligotrophic, mesotrophic lake with thickets of macrophytes.

Zone 4 (depth 275–245 cm) reflects a significant decrease in the level and swamping of the lake 2110–1760 cal yr BP. *Aulacoseira laevissima*, typical for oligotrophic water bodies, became dominant. This species occurs in shallow glacial lakes (~1 m deep) in the subalpine and alpine belts [34,46]. *A. laevissima* produces an outbreak ~1990–1930 cal yr BP, resulting in a severe decrease in diatom diversity. The content of benthic species increased, especially *Pinnularia crucifera*. The subdominants are the epiphytes of oligotrophic and oligotrophic dystrophic lakes. The arctoboreal *Encyonopsis amphioxys*, the cosmopolitans *Eunotia glacialis* and *Tabellaria flocculosa*, and the arctoboreal *Eunotia lapponica* and *Eunotia serra* that often inhabit wet sphagnum mosses appeared. There was an increase of acidophiles (up to 25.2%) and halophobes (up to 26.4%), typical for bog environments, and arctoboreal diatoms (12–25.5%). The concentration of valves decreased (1.4–9.3 × 10^6^ valves/g). Significant fluctuations in the diatom proportions are reflected in the variations of the PC1 and PC2 axes of indicator species (Figure 4).

Zone 5 (1760–1130 cal yr BP, depth 245–180 cm) indicates that the hydrological regime of the oligotrophic dystrophic reservoir was unstable. The content of planktonic species varied greatly; *Aulacoseira laevissima, A. italica, A. crenulata,* and *A. subarctica* were found. The content of *Tabellaria flocculosa*, which is typical for peat bogs, increased significantly. Halophobes and acidophiles *Eunotia serra* and *E. lapponica*, which develop optimally at pH 4.9, became permanent components of the flora [30]. The maximum content (up to 10.8%) of these species was ~1500–1440 cal yr BP. At that time, the proportion of planktonic species sharply reduced (up to 1% of the assemblage); the lake may have been heavily overgrown for a short period. The proportion of arctoboreal diatoms rose to 33%, while their content in the underlying and overlying sediments was 15–26.7%. The abundance of valves does not exceed 2.5 × 10^6^ valves/g.

Zone 6 (depth 180–145 cm) reflects a short-term fall in the lake water level ~1130–940 cal yr BP. The role of *Pinnularia crucifera* increased among the dominants. The benthic *Pinnularia genkalii* appeared among the subdominants. The content of bog species of the genus *Eunotia* increased (up to 27%), including *E. paludosa*, which is typical of sphagnum bogs and tolerant to temporary dry environments [47]. The concentration of valves varies within 0.7–1.8 × 10^6^ valves/g.

A significant drop in the lake level, as a result of further overgrowth, was about 940–760 cal yr BP (zone 7, depth 145–120 cm). The content of planktonic diatoms decreased, and benthic species and epiphytes predominated an increase in aquatic macrophytes. The proportion of species of the genera *Eunotia* (up to 35.5%) and *Pinnularia* (up to 45.8%) increased, among which species characteristic of near-neutral or slightly acidic waters of oligotrophic reservoirs dominate (*Eunotia glacialis, Pinnularia crucifera, P. genkalii*). Subdominants are *Encyonopsis amphioxys* and *Tabbelaria flocculosa*. In the periods 910–870 and 840–800 cal yr BP (depths 140–135 and 130–125 cm), complete overgrowth of the lake is recorded. The concentration of valves decreased to 0.2 × 10^6^ valves/g. At the beginning of this stage, when the lake was completely overgrown, diatom diversity decreased sharply.

Zone 8 (depth 120–105 cm) reflects a slight inundation of the lake ~760–660 cal yr BP. The content of *Aulacoseira laevissima* increased, and the proportion of bottom species decreased. Climatic conditions were cold; the content of arctoboreal species, including *Eunotia serra,* reached 29.5%. The concentration of valves increased to 2.1 × 10^6^ valves/g.

Stronger watering due to increased precipitation was recorded at ~660–250 cal yr BP (zone 9, depth 105–40 cm). The proportion of planktonic species increased to 49.7%, indicating the watering of the catchment. *Aulacoseira laevissima* became abundant. Climatic conditions remained cold, and the content of arctoboreal species reached up to 29%. The concentration of valves varied from 1 to 4.7 × 10^6^ valves/g; diatom productivity was unstable. Short-term shallowing of the lake was recorded ~380–350 cal yr BP (depth 60–55 cm). Here, the content of planktonic species decreased to 8.1%, and the participation of *Eunotia paludosa* and *E. nymanniana*, which are able to live with insignificant moisture [47], increased. The diversity of diatoms possibly increased due to the formation of various biotopes during seasons with contrast moisture. The variations of the PC1 and PC2 axes show the instability of environments and lake ecosystems (Figure 4).

The subsequent development of the lake took place under conditions of progressive shallowing. Zone 10 (depth 40–0 cm) with the domination of epiphytes (up to 77%), such as *Eunotia paludosa* and *E. glacialis*, and subdominants *Eunotia nymanniana* and *Encyonema paucistriatum* indicates weak moisture. In the top layer, the content of benthic *Navicula angusta*, which is distributed mainly in oligotrophic waters and mosses in mountainous regions, increased [33]. An increase in the content of planktonic species (*Aulacoseira laevissima, A. italica, A. subarctica, A. crenulata*) up to 25.4% shows significant watering of the lake in the first half of the 19th century. The content of *Tabellaria flocculosa* also increased significantly. Less severe inundation of the lake manifested in the first half of the 20th century—the proportion of planktonic diatoms in-depth 0.10–0.15 m reaches 6.3%. The decrease of planktonic species in the top (1.2%) indicates complete overgrowth of the lake. In peat formed in the phases of watering, the concentration of valves reached 3.5 × 10^6^ valves/g; in other layers, it was less than 2 × 10^6^ valves/g. In general, acidophiles and halophobes dominated. The content of arctoboreal species decreased from 32.7 to 10% in the top.

## 5. Discussion

Previous studies provided evidence of the development of the small lacustrine system in relation to environmental variations, and showed high sensitivity to the mountain region of the Far East [14,18,19,20,21,44,48]. The study of two sections of organic deposits in lake basins showed that large landslides on the slopes of the Solontsovsky paleovolcano took place repeatedly, resulting in the formation of multiple small lakes. Izyubrinye Solontsi Lake (4400 years) is older than Nizhnee Lake (2600 years); the course of their development was metachronous (Figure 6), most likely due to different size, depth, and altitudinal positions, but the trends were similar and reflect the climatic changes in the late Holocene in the Central Sikhote-Alin Mountains.

Figure 7 illustrates a comparison of our results for paleoclimate proxies with sun activity [7] and palaeotemperatures [49,50]. The main factor determining the development of the lakes and sedimentation was the change in moisture, which was controlled by the precipitation. Mire vegetation and diatom microflora in the lakes were highly dependent on changes in watering, as evidenced by the frequent change of peat-forming plants with different trophic preferences, and diatom assemblages with species that prefer different habitats and geochemical environments. The cover of vascular plants in the mires very weakly reacted to climatic changes with two or three species of sedges and cottongrasses, with some hydrophytes dominating throughout the entire period.

Due to substantial seasonal fluctuations in the level of swamp waters inherent in this region, saturation periodically changed from stagnant, accompanied by an outbreak of green mosses development, to weakly flowing, as indicated by periodically appearing terrigenous material. Significant seasonal fluctuations in the water level were typical for the entire period.

The formation of the Nizhnee Lake and the beginning of peat accumulation occurred during a global cool interval (2800–2600 cal yr BP) and was accompanied by a decrease in moisture in Asia [51]. The intensity of summer monsoon weakened at that time (3600–2100 cal yr BP) in Northeast China [2,3,52]. In the Lower Amur region, the climate was especially dry at 2570 cal yr BP [12]. A moisture decrease was recorded at 2735–2040 cal yr BP in the Muta peat bog (570 m a.s.l.), located on the main watershed [53]. A progressive decrease in depth of the paleolake was noted at 3010–2630 cal yr BP for the Larchenkovo swamp of Shkotovskoe Plateau (730 m a.s.l.) [19].

At the initial stage, the Nizhnee Lake was a mesotrophic eutrophic lake, more watered than Izyubrinye Solontsi Lake. The flooding of the basins depended on local geomorphological features. The water in the Nizhnee Lake was weakly alkaline, apparently, due to the parent substrate. The period 2590–2430 cal yr BP was characterized by prolonged dry seasons and forest fires, as suggested by charcoal in the peat. Activation of fires at that time was also recorded in peat bog sections of the Bikin River basin [54] and the Southern Sikhote-Alin [55].

The drop of the Nizhnee Lake level was recorded during a short-term cooling of 2330–2280 cal yr BP. Overgrowing of the lake led to a decrease in pH value. The productivity of diatoms decreased. The abundance of halophobes indicates an increase in atmospheric nutrition. The decrease in lake level and infilling by the vegetation of Izyubrinye Solontsi Lake began ~ 2270 years ago; peat with *Larix* accumulated around the lake [22]. At that time, species capable of living in a low moisture environment were widely developed in the diatom flora. The benthic *Pinnularia borealis*, and the acidophilic *Eunotia praerupta* dominated among epiphytes in the Izyubrinye Solontsi Lake; *Hantzschia amphioxys* dominated in the Nizhnee Lake. According to estimates made for the seacoast (Langou I Bay), temperatures reached a minimum (about 1 °C below the present) ~2280 cal yr BP [18]. Severe frosts and snowfalls in the 3rd–4th centuries BC are noted in records from China [50].

The maximum lake level of the Nizhnee Lake was ~2280–2110 cal yr BP, which coincides with slight warming connected with increased solar activity [7,49]. There was an increase in diatoms in the lake, especially planktonic species. The pH values were close to neutral. The sediment content of peat increased due to erosion during frequent heavy rains associated with the passage of typhoons [54]. It is possible that a lack of vegetation on landslide surfaces facilitated erosion and transport of sediment into the lake. It is also possible that a new landslide formed in the upper part of the catchment above the Nizhnee Lake at this time. An abundance of *Aulacoseira subarctica* (Figure 6) may indicate increased snowfall and thick ice cover [44]. This species is capable of developing under low light conditions. A sharp increase in its proportion possibly indicates the activation of erosion during snowmelt or typhoons. The activation of winter cyclogenesis occurred under the conditions of weakening of the Siberian High [56,57]. The weakening of the Siberian anticyclone contributed to the activation of winter cyclogenesis. Sphagnum and green mosses disappeared from the swamp, and herbaceaus peat accumulated. Active slope wash and increased terrigenous supply were possible reasons for the reduction of the moss cover. Environmental changes in Southern Primorye indicate weak summer monsoon at this time [14]; studied lakes located in northernmost part of the monsoon region. We suggest that more active cyclogenesis occurred at this time in this area. The increased precipitation ~2370–2150 cal yr BP was recorded on the mires of the western macroslope of the Central Sikhote-Alin (Bikin River basin) and to the south, on the Sergeev Plateau mires (2380–2130 cal yr BP) [20,54]. On the eastern macroslope of the Sikhote-Alin, the maximum flooding of the swamp in the valley of the Milogradovka River was 2310–2250 cal yr BP [21].

A gradual fall of the lake level with some fluctuations took place in cooler conditions ~2110–1760 cal yr BP. The drop in annual temperature was ~1 °C lower than the present [18]. Cladocera appeared in high abundance in the Nizhnee Lake. At that time, the green mosses *Limprichtia revolvens* and *Scorpidium scorpioides*, typical for waterlogged swamp areas [58], developed widely on the mire. The peak of planktonic diatoms indicates a short-term watering ~1990–1930 cal yr BP. The peak of *Aulacoseira laevissima*, usually living in shallow glacial lakes, indicates a cold event correlated with short-time solar minima [7,49] (Figure 6). Testate amoebae, typical of heavily watered environments, including sphagnophiles, developed in abundance ~1870 cal yr BP. A sharp change in diatom taxonomic composition indicates significant transformations in the lake ecosystem.

Unlike Izyubrinye Solontsi Lake, which was relatively stable 2270–1230 cal yr BP with a tendency of shallowing and desiccation, the Nizhnee Lake had an unstable regime between 1761 and 1133 cal yr BP (Figure 6). The geochemical environment changed ~1710 cal yr BP; the pH value decreased (to 4.9). Cold conditions are indicated by the development of a stable sphagnum cover (1870–1550, 1390–1340 cal yr BP). The peak of cooling in the Central Sikhote-Alin was accompanied by a significant reduction of atmospheric precipitation and caused a short period of strong overgrowth of the lake ~1500–1440 cal yr BP. It is correlated with the end of the global cold event (1650–1450 cal yr BP), which was accompanied by drying in Asia [51,52,56]. At this time (1600–1300 cal yr BP), a weakening of the summer monsoon was recorded [2]. Some authors consider the boundaries of this cooling in a wider range—1750–1350 cal yr BP [49]. The cooling manifested in Primorye [59] and Sakhalin [13], cold Kofun stage (1760–1220 cal yr BP) was identified in the Japanese Islands [9]. A short-term cooling and strong mountain lake shallowing were recorded between 1800 and 1500 cal yr BP on Kamchatka [48,60].

The subsequent development of the Nizhnee Lake reflects the unstable climatic variability during the transition to the Medieval Warm Period. A short-term cooling between ~1290 and 1260 cal yr BP led to the development of sphagnum mosses on the mire near the lake. Perhaps, due to the weak decomposition of sphagnum remains, the rates of peat accumulation increased. Development of green moss *Limprichtia revolvens* and *Baeothryon* indicates more cold wet environments with stagnant water regimes 1260–1200 cal yr BP. Hypnum peat accumulated at the time. The watering phase of the Izyubrinye Solontsi Lake was also noted at ~1230 cal yr BP [22]. A long period with abundant river flow and severe floods in the Bikin River basin began ~1260 cal yr BP [54]. This cold event coincided with solar minima 1300–1200 cal yr BP [7,56].

The warming correlated with the Medieval Warm Period is identified by pollen data from Izyubrinye Solontsi Lake sediments ~1080–810 cal yr BP [22]. On the seacoast of eastern Primorye annual temperature was ~1.5 °C higher than present [18]. Nizhnee Lake water level decreased ~1130–940 cal yr BP. In the mire, the role of herbaceous plants increased. Among the mosses, *Sphagnum divinum*, typical for mesotrophic bogs, developed. The role of green mosses diminished sharply between ~1010 and 980 cal yr BP. The presence of microcharcoal indicates fires in the surrounding territories (1130–1100, 1070–1040, 1010–980 cal yr BP). Frequent fires during the Medieval Warm Period also occurred in the South Sikhote-Alin, possibly connected with human activity in the Middle Ages [20].

The lake level fall and active overgrowth of the Nizhnee Lake ~940–760 cal yr BP probably occurred during a temperature decrease. The lake may have been completely overgrown 910–870 and 840–800 cal yr BP. The Izyubrinye Solontsi Lake reduced 960–840 cal yr BP. Here, a cold episode is recorded ~840–810 cal yr BP; there was a small fire near the lake.

The increase in Nizhnee Lake water level began with a change in the climatic regime 760–660 cal yr BP and especially during 660–250 cal yr BP (the Little Ice Age). The increase in precipitation resulted in a deepening of Izyubrinye Solontsi Lake. This cooling is related to solar forcing—solar activity decreased significantly [7,61,62]. The annual temperature was 1.5–2 °C lower than present [50,63,64]. The Little Ice Age was likely the coldest period of the last 8000 years; the climate was unstable with high variability of seasonal temperatures [61,62]. The climatic condition of the southern Far East was controlled by the strengthening of the Siberian High and the Aleutian Low [65]. Apparently, at that time, there was an increase in cyclonic activity. Summer monsoons became active with a short-time period of weakening [2,66]. Winter monsoon was weak after 1000 cal yr BP and intensified during the Little Ice Age [8,67].

The mire vegetation and diatom communities of the Solontsovskie lakes responded to highly variable environmental changes during the Little Ice Age. In the Central Sikhote-Alin, conditions in the first half of this period were colder, as in other continental regions of the southern Far East [54,68]. The proportion of arctoboreal diatoms and *Aulacoseira laevissima* increased, especially 610–420 cal yr BP. The conditions were cold and wet; sphagnum mosses were widespread in the mire. The development of the green moss *Meesia trifaria* on the margin of the Nizhnee Lake indicates an increase in mineral nutrition.

The decrease in the proportion of arctoboreal species is an indicator of several short-term warmer episodes (660–620, 530–500, 440–410 cal yr BP). As a rule, the proportion of phytoplankton increased at this time. The abundance peaks of arctoboreal diatoms (690–660, 590–560, 470–440 cal yr BP) apparently correspond to minimums of solar activity, including Wolf and Shpörer minima [61,62,69]. The number of cladocera was sharply reduced from 590 years ago; they disappeared with the further overgrowing of the lake. On the western slope of Sikhote Alin, flooding of the valleys and an increase in the frequency of floods were observed at 645–550 and 490–420 cal yr BP; the cold, dry phase was ~420–220 cal yr BP [54]. Short-term shallowing of the Nizhnee Lake is recorded at 380–350 cal yr BP; the abundance of arctoboreal diatoms increased, the grass layer on the mire became more developed, and the participation of sphagnum mosses decreased sharply. At the time, a cold episode stands out in the global records [51,61]. Charcoal pieces indicate a fire near the lake ~350–290 cal yr BP. The mineralization of peat increased somewhat, and the swamp became drier.

Over the last 250 years, the lake has progressively become shallower due to a common tendency of development. The hydrological conditions in the basin became unstable, and the pH decreased. The Maunder minimum (1645–1710 CE) is documented by a decline in watering and an abundance of arctoboreal diatoms between ~240 and 210 cal yr BP. The number and variety of green mosses increased in the mire. After the lake became shallower, an increasing abundance of planktonic species indicated flooding occurred in the first half of the 19th century, and a less significant one at the beginning of the 20th century.

## 6. Conclusions

The paleoenvironmental records from small lakes indicated several short-term Late Holocene climatic events in the mountain region in the Russian southern Far East. The sediments of the Solontsovskie lakes are natural high-resolution archives that allow us to restore a detailed record of the paleoclimatic events. A comparison of small lake data showed that the Nizhnee Lake was more sensitive to hydroclimatic changes and showed high environmental variability. Izyubrinye Solontsi Lake was more stable, but both lakes had similar general trends. Climate instability in the last 2600 years determined the features of the hydrological conditions of the Nizhnee Lake, which were expressed in frequent changes in diatom assemblages and peat-forming plants. A lake with a mesotrophic eutrophic regime became oligotrophic, mesotrophic at 2330 cal yr BP; its maximum depth and productivity were between 2280 and 2110 cal yr BP. It became oligotrophic at 2110 cal yr BP, and from 1760 cal yr BP, oligotrophic dystrophic. For a long period, the moistening conditions were unstable, and the stages of watering and drainage, up to short-term episodes of complete overgrowth, were distinguished. A decrease in the lake level was significant over the last millennium. The lake level especially dropped in the last 250 years; pH decreased, and a flooded oligotrophic dystrophic mire developed near the lake. Multiple succession events in sphagnum mosses with different trophic preferences indicate that the mire vegetation was unstable. Drying phases, as a rule, corresponded to cooling periods. Climate fluctuations appear to be linked to the intensity of summer monsoon. The Little Ice Age was the exception, and was characterized by high moisture and an increase in precipitation that was connected with active cyclogenesis. The signals of solar minima are traced in high-resolution proxy records. High variability of the lake ecosystem demonstrated changes in moisture availability within the atmosphere–ocean–land system, which resulted in a regional and microclimate response to global climatic events, variability of modes of atmospheric circulation, and changes in predominant cyclone tracks.

## Figures and Tables

**Figure 1 biology-12-00913-f001:**
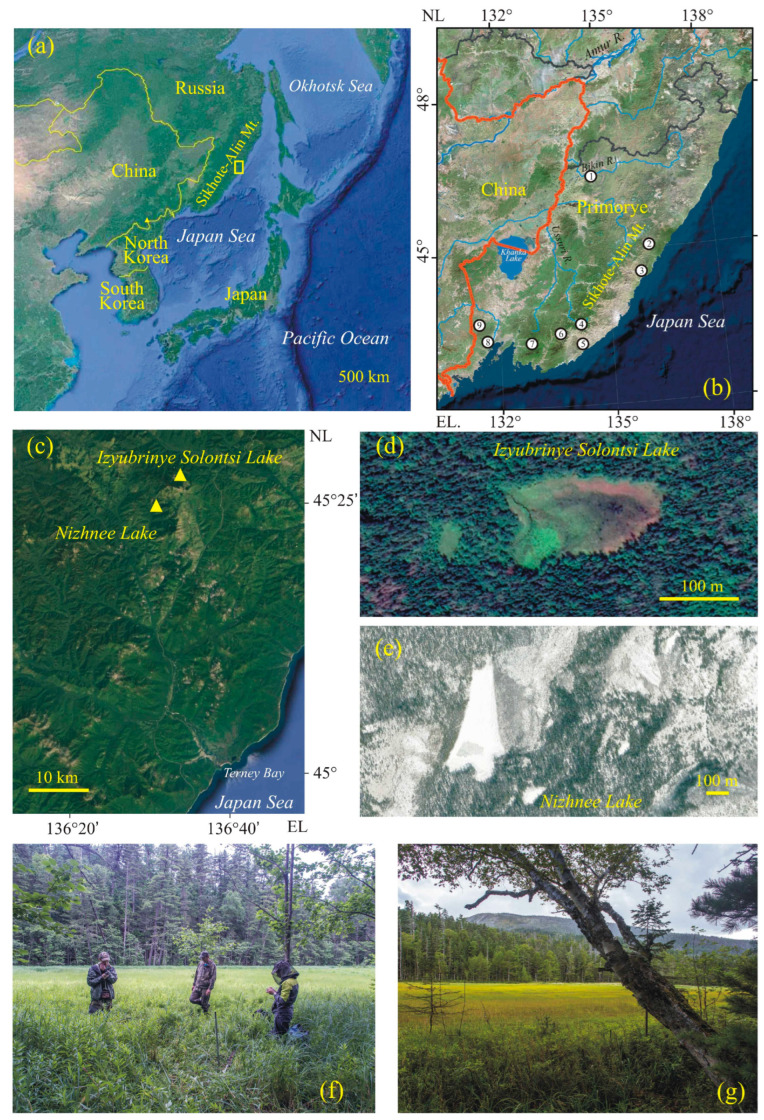
Study area: (**a**) Position of the study area in NW Pacific region; (**b**) 1—Bikin River; 2—Solontsovskie lakes; 3—Langou I Bay; 4—Muta peat bog; 5—swamp in Milogradovka River valley; 6—paleolakes of Sergeev Plateau; 7—Larchenkovo swamp of Shkotovskoe Plateau; 8—Utinoe Lake; 9–paleolake of Shufan Plateau; (**c**) Solontsovskie lakes location and position of the studied sections; (**d**,**g**) Nizhnee Lake; and (**e**,**f**) Izyubrinye Solontsi Lake.

**Figure 2 biology-12-00913-f002:**
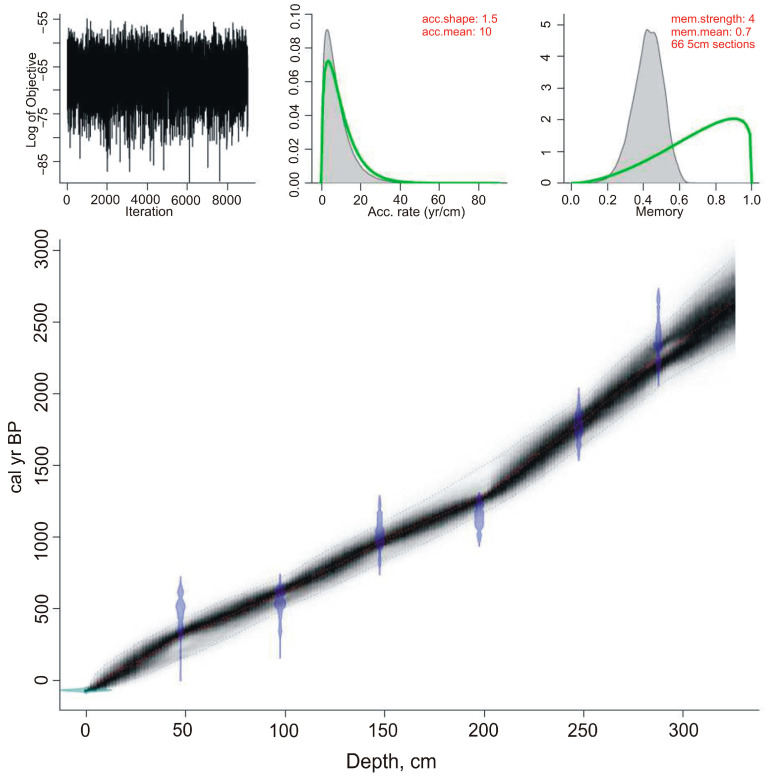
Age–depth model for the studied Section 0315 from the Nizhnee Lake sediments, plotted using Bacon [39] with 95% confidence limits shown.

**Figure 3 biology-12-00913-f003:**
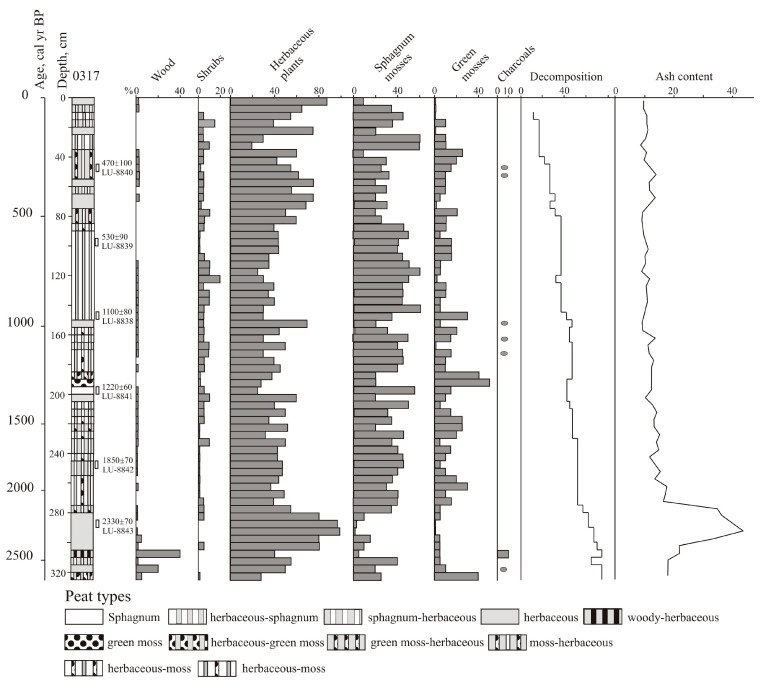
Botanical composition and characteristics in Section 0315, the Nizhnee Lake, plus records of charcoal and terrigenous input; Oval shapes show presence of charcoal.

**Figure 4 biology-12-00913-f004:**
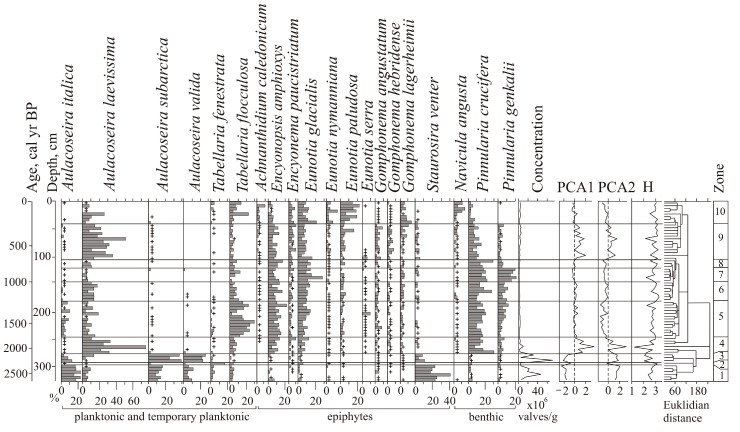
Diatom percentage diagram showing the distribution of the diatom taxa, with a relative abundance above 5% in a sediment core from the Nizhnee Lake (site 0317), PCA axes 1 and 2 scores for diatom data, and Shannon index H. A ‘+’ sign indicates the content of frustules less than 1%.

**Figure 5 biology-12-00913-f005:**
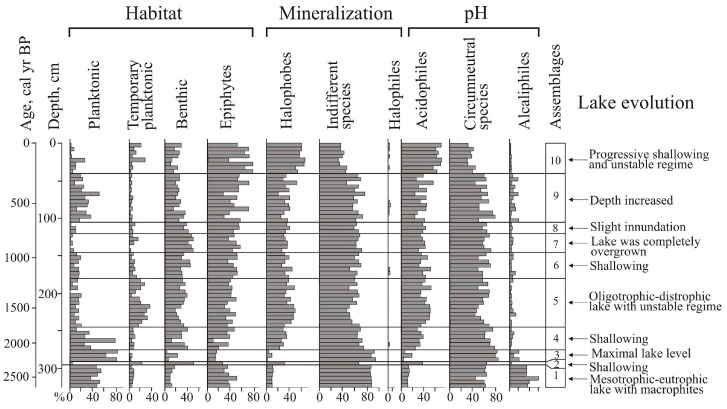
Distribution of ecological groups of diatoms in a sediment core from the Nizhnee Lake (site 0317) with regard to habitat type, salinity, pH, and geographical distribution.

**Figure 6 biology-12-00913-f006:**
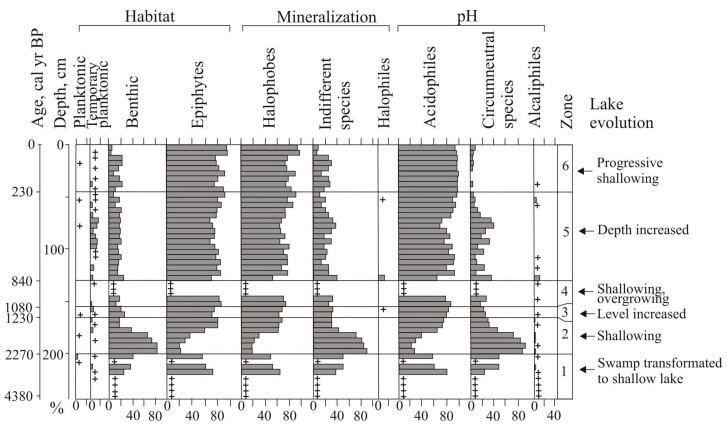
Distribution of ecological groups of diatoms in a sediment core from the Izyubrinye Solontsi Lake (site 7115) with regard to habitat type, salinity, pH, and geographical distribution, a ‘+’ sign indicates singe diatom valves.

**Figure 7 biology-12-00913-f007:**
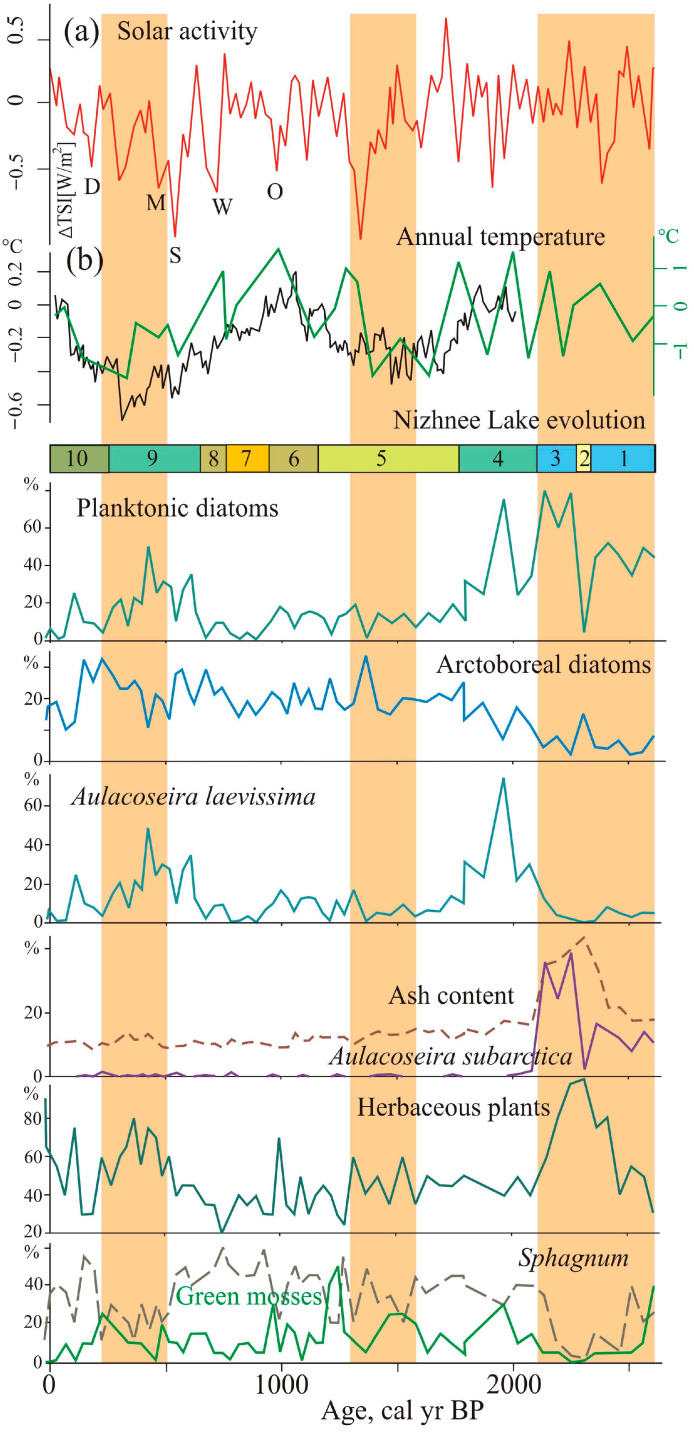
Compilation of selected proxy records from sediments of the Nizhnee Lake with solar activity and palaeotemperatures. (**a**) Solar activity fluctuations reconstructed based on 10Be measurements in polar ice [7], periods of the Grand Solar Minima: O—Oort, W—Wolf, S—Spörer, M—Maunder, D—Dalton; (**b**) the black line shows Decadal mean temperature variations (°C) during AD 1961–1990, estimations for extratropical Northern Hemisphere (90–30° N) [49]; the green line—reconstruction of annual temperatures for the Amur River Region from modern average temperature [50]. The vertical orange bars show the abrupt summer monsoon declines [2,3].

**Table 1 biology-12-00913-t001:** Radiocarbon dates and accumulation rates obtained for sediments of the Nizhnee Lake, Sikhote-Alin Mountains.

Lab Number, LU-	Sample Number	Depth, cm	^14^C-Age, yr BP	Calendar Age (2σ)	Sedimentation Rate, mm/yr
8838	10/0317	45–50	470 ± 100	480 ± 100	1.2–1.4
8839	20/0317	95–100	530 ± 90	550 ± 80	1.6–1.7
8840	30/0317	145–150	1100 ± 80	1030 ± 90	1.4–1.5
8841	40/0317	195–200	1220 ± 60	1150 ± 70	1.6–1.7
8842	50/0317	245–250	1850 ± 70	1780 ± 90	1–1.2
8843	58/0317	285–290	2330 ± 70	2380 ± 130	0.8–1

## Data Availability

The data are available on request from the authors.

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
