# Peer review of "High-Resolution Lacustrine Records of the Late Holocene Hydroclimate of the Sikhote-Alin Mountains, Russian Far East"

_biology, 2023, doi:10.3390/biology12070913_

Round 1

Reviewer 1 Report

This is an interesting an valuable manuscript. Not much has been written on the climate-driven transitions between lake and swamp or mire conditons. Most of my comments are on the manuscript. The manuscript would would benefit from a more complete description of the core and the changes that took place in the diatom flora. Because one major point is to identify how climate affects these lakes 1) in what major time intervals do both lakes exhibit evidence of climate change and in what intervals do they differ, 2) how large are the differences. As a diatomist and paleolimnologist it is difficult to verify the scientific soundness without a more complete description of the changes in the diatom floral changes.

It is clear that English is not the first language of any of the authors a fair amount of editing is necessary to ensure that the author's meaning is properly translated into English.

Author Response

Please see the attechment

Reviewer 2 Report

Overview and general recommendation:

This manuscript focuses on changes that occurred in small local mountain lakes in the Russian Far East, the approach and methodology of the work are adequate. Nevertheless, despite the fact that I consider the work interesting, I feel that the quality and approach of the manuscript do not reach what is expected in publications of this journal. The first problem I see is the scope of the work, I consider that it is too local; perhaps it would fit better in a more specific magazine or journal focused on the study region.

On the other hand, many figures need to be checked; some are clearly neglected (e.g., Figure 1 and Figure 4), with evident problemns of presentation that sometimes make it difficult to interpret the data. On the other hand, the discussion of the data is not entirely clear and sometimes falls into overinterpretation, which could be avoided with more detailed explanations.

Other Small Suggestions:

Line 17: “heat-loving” andcold-loving elements” This is a really unspecific expression. Please rewrite.

Line 18: “rare animals live in this area, including the Amur tiger” Is this study relevant in any way for the Amur tiger?, I don’t think so, I suggest to delete this sentence.

Line 29: ecological and taxonomical composition of…

Line 46-48: weird sentence, I suggest rewriting.

Line 49-50: Please add some references in this sentence.

Line 55: “During the Holocene” instead of “in the Holocene”

Line 61: “In the Southern Russian Far East observations began in the second half of the 20th century” which kind of observations? Please be more specific…

Line 68: “an important role in changing areals”, I guess you want to say areas instead of areals. Please check.

Line 71: “largely determined” I think that “determined mainly” fits better here.

Figure 1: Tthis figure does not show the clear positions of the studied lakes. A more specific map is needed. In addition, its presentation is not adequate for a jorunal like this. Redo the figure with more explanatory images and take care of details such as the position of the letters and scales.

Line 116: “Sphagnum mosses (10–35%)” What do these % mean? Abundance? If so, why don't you show the percentages for the other botanical groups that are cited in the same sentence? Please explain this and show all the percentages.

Line 119: “belongs to the mesotrophic type” please explain this or add reference.

Figure 2: The figure is barely visible, especially the upper graphs, I suggest rearranging the figure or dividing it into several figures to facilitate the interpretation of the data for the reader.

Line 186: “lily Nymphaéa álba” Scientific names are in Latin, so please delete the diacritical marks: Nymphaea alba.

Line : “144 taxa of diatoms were identified “ When the authors say taxa, do they refer to Genera? Families? Species? or all level of taxa together? Specify more please...

Figure 4: This figure is really difficult to interpretate, because different axes and graphics are not adequately orientated. Please check this figure and correct this.

Figure caption 4: “showing the distribution of the main diatom taxa in sediment”. What criteria is used to determine which diatom taxa are "main", what relative percentage of abundance has been used? it may be good to indicate it in the material and methods...

Line 118: “Green moss is present singly” What do you mean with this?

Line 320: “ Aulacoseira laevissima became dominant indicating the watering of the catchment What do you mean with dominant? more abundant? This is too ambiguous... Also the assumption that this “dominance” indicates “the watering of the catchment” is a bit over interpretative, I don't think this is the only possible interpretation. This needs to be better discussed.

Line 321: “Climatic conditions remained cold” Which data support this sentence? specify it in the text please, so the reader does not have to be stopping to review the graphs / literature to verify what is argued...

The writing is not bad, but some sentences sound unnatural or difficult to understand. I suggest a deep review of the text.

Reviewer 3 Report

The authors reported the high-resolution botanical and diatom records from Nizhnee Lake, and ten stages reflecting watering and swamping of the lake were identified. The main point of the manuscript is that the hydrological variations of the Nizhnee Lake were controlled by the precipitation changes, which is related to the intensity of summer monsoon modulated by the solar activities. However, the quality of the manuscript should be improved before publications, because there are still some errors and reasoning and language problems in the current version.

1.  Most importantly, the authors should make it clear about their reasoning. I am lost after reading the discussion part, as sometimes the overgrowth of the lake corresponds to warm periods and sometimes not. So, more explanations are necessary for the readers.

2.  Actually, only the data from Nizhnee Lake is reported, but the authors mentioned the results of Solontsovskie Lake throughout the manuscript (e.g., in the abstract, figure 1 discussion etc..) without showing any data. The author should provide the comparisons of the two lakes in a separate figure.

3.  More information of radiocarbon data should be provided, for example, what materials are used and how the authors deal with the old carbon effect?

4.  See more line-by-line comments highlighted in the attached file.

See details in the attached file.
